# The risk of iatrogenic radial nerve and/or profunda brachii artery injury in anterolateral humeral plating using a 4.5 mm narrow DCP: A cadaveric study

Chaiwat Chuaychoosakoon⊕, Supatat Chirattikalwong⊕, Watit Wuttimanop‡, Tanarat Boonriong‡, Wachiraphan Parinyakhup‡, Sitthiphong Suwannaphisit[ID]⊕*

Department of Orthopedics, Faculty of Medicine, Prince of Songkla University, Hat Yai, Songkhla, Thailand

⊕ These authors contributed equally to this work.
‡ These authors also contributed equally to this work.
* psu.orthopedics@gmail.com, aunsittipong@gmail.com

## Abstract

### Introduction

Fixation of humeral shaft fractures with a plate and screws can endanger the neurovascular structure if proper care is not taken. No studies to our knowledge have studied the risk of iatrogenic radial nerve and/or profunda brachii artery (RNPBA) injury from each screw hole of a 4.5 mm narrow dynamic compression plate (narrow DCP). The purpose of this study is to evaluate the risk of RNPBA injury in anterolateral humeral plating with a 4.5 mm narrow DCP.

### Material and methods

18 humeri of 9 fresh-frozen cadavers in the supine position were exposed via the anterolateral approach with 45 degrees of arm abduction. A hypothetical fracture line was marked at the midpoint of each humerus. A precontoured ten-hole 4.5mm narrow DCP was applied to the anterolateral surface of the humerus using the fracture line to position the center of the plate. All screw holes were drilled and screws inserted. The cadaver was then turned over to the prone position with 45 degrees of arm abduction, and the RNPBA exposed. The holes through in which 100% of the screw had contact with or penetrated the RNPBA were identified as dangerous screw holes, while lesser percentages of contact were defined as risky.

### Results

The relative distance ratios of the entire humeral length from the lateral epicondyle of the humerus to the 4th, 3rd, 2nd and 1st proximal holes were 0.64, 0.60, 0.56 and 0.52, respectively. The most dangerous screw hole was the 2nd proximal, in which all 18 screws had contacted or penetrated the nerve, followed by the risky 1st (12/18), 3rd (8/18) and 4th (2/18) holes.

**Data Availability Statement:** All relevant data are within the manuscript and its Supporting Information files.

**Funding:** Initials of the authors who received each award (C.C.) Grant numbers awarded to each author (REC 62-434-11-1) The full name of each funder (Faculty of Medicine, Prince of Songkla University) Did the sponsors or funders play any role in the study design, data collection and analysis, decision to publish, or preparation of the manuscript? No, the funders had no role in study design, data collection and analysis, decision to publish, or preparation of the manuscript.

**Competing interests:** The authors have declared that no competing interests exist.

**Abbreviations:** DCP, dynamic compression plate; RBPBA, radial nerve and/or profunda brachii artery; LCP, locking compression plate.

## Conclusion

In humeral shaft plating with the 4.5mm narrow DCP using the anterolateral approach, the 2nd proximal screw hole carries the highest risk of iatrogenic radial nerve and/or profunda brachii artery injury.

## Introduction

In a humeral shaft fracture, the middle third of the diaphysis was the most common fracture location with the incidence of 60% [1]. The radial nerve and profunda brachii artery could have been injured during the trauma that caused the fracture, or they could be injured during the fixation from drilling the screw holes or inserting the screws [2–5], as these structures are located on the posterior aspect of the humeral shaft, crossing the humerus obliquely from the posteromedial to the posterolateral sides, but their precise location cannot be known by the surgeon performing the fixation and drilling screw holes and inserting screws [6–8]. Earlier studies have reported incidences of radial nerve injury during surgical fixation ranging from 3.57% to 18.20% [9–11].

There are two main treatment methods for a humeral shaft fracture. The first is conservative treatment with a "sugar tong splint" or "functional brace", and the second is operative treatment with a plate or nail system [12, 13]. In mid-shaft humeral fixation, humeral shaft plating is preferred by most surgeons to humeral shaft nailing [14, 15], with some studies showing higher functional outcomes [16] and lower complication rates [17–19] when fixing with plate and screw. Some surgeons prefer to do humeral shaft plating using the anterolateral approach while some surgeons prefer the posterior approach [14]. The posterior plating is more suitable for mid- to distal shaft humerus fractures, while the anterolateral plating is more suitable for proximal to mid-shaft humerus fractures. There have been three studies comparing the anterolateral and posterior plating in humeral mid-shaft fixation [9–11], in all of which both plating had excellent outcomes in union rate, range of shoulder and elbow motions, and functional scores. The complication of iatrogenic radial nerve palsy in most studies were similar between the two locations of the plate which the incidence were 3.57–14.20% in the anterolateral plating and 16.67–18.18% using the posterior plating. In contrast, one study reported that the incidence of iatrogenic radial nerve palsy in the anterolateral approach was higher than the posterior approach. Wang et al. reported on 39 cases of iatrogenic radial nerve injury during plating, 37 cases of anterolateral plating and 2 cases of posterior plating [4].

In the anterolateral plating, the surgeon can use one of two common types of plate, the 4.5 mm narrow dynamic compression plate (narrow DCP) or the 4.5 mm narrow locking compression plate (narrow LCP) [10, 20–23]. These plates are applied to the humerus in different locations, and thus the trajectory of the screws is also different. The 4.5 narrow DCP is commonly applied at the anterolateral surface of the humerus (Fig 1) [9–11, 24], while the 4.5 mm narrow LCP can be applied at either the mid-anterior aspect (Fig 2) or the anterolateral surface [21, 25, 26]. The direction of the screws while applying the plate at the anterolateral surface is from the anterolateral aspect of the humerus to the posteromedial aspect, while the direction of the screws while applying the plate at the mid-anterior aspect of the humerus is from the mid-anterior to the mid-posterior area of the humerus. The radial nerve and profunda brachii artery run from the posteromedial to the posterolateral aspect of the humerus, and are at risk of iatrogenic injury in both anterolateral and posterior plating. When applying the plate at the anterolateral surface of the bone, earlier studies have reported a risk while drilling some screw

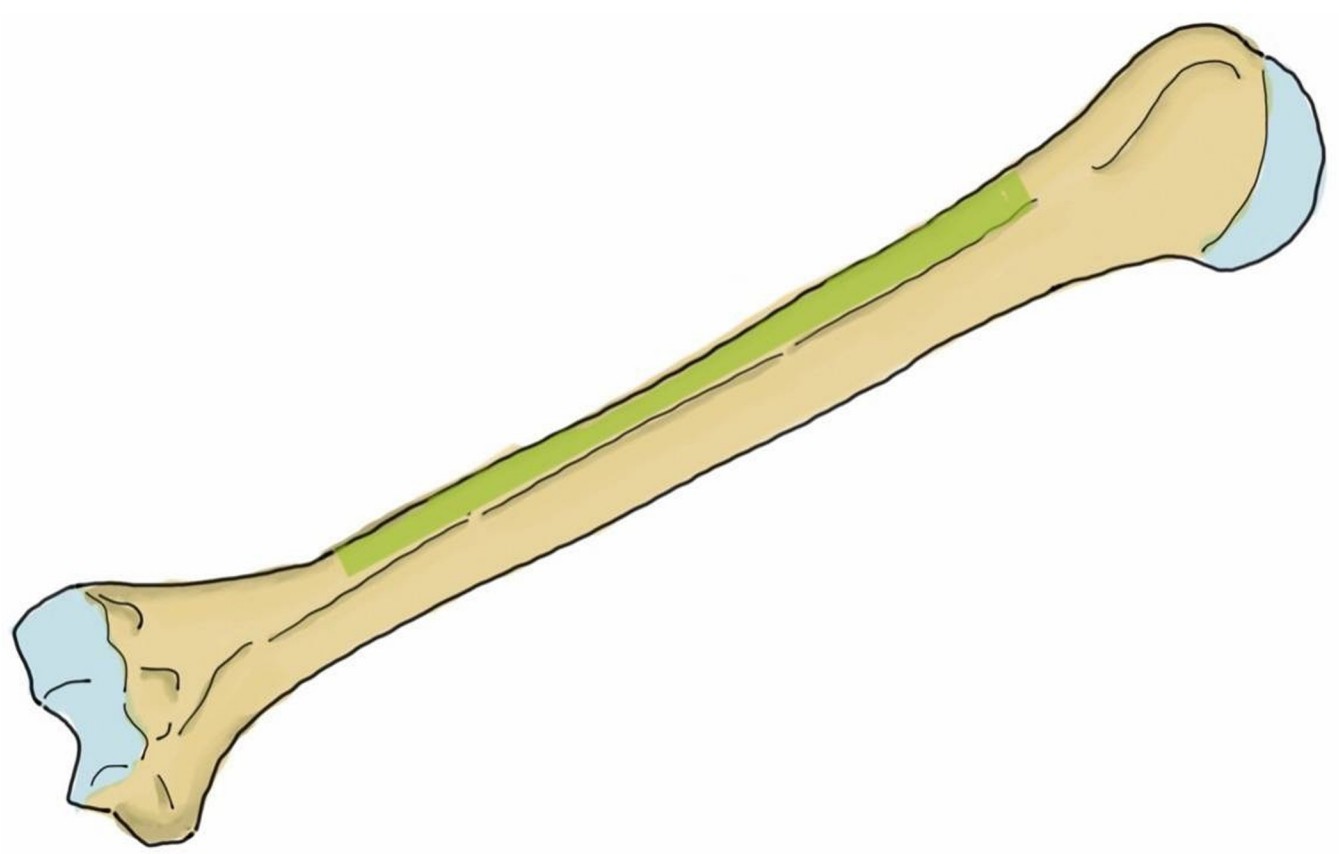

**Fig 1. The anterolateral surface of the humerus (green shaded area) shows the area in which the plate can be applied.**

holes and/or inserting the screws but none of these studies provided the individual risk of each hole position. When applying the plate at the posterior aspect of the humerus, earlier studies have reported a risk in the step of finding the radial nerve and profunda brachii artery as they pass across the mid-posterior aspect of the humerus before applying the plate [6–8]. A study by Apivatthakakul et al. evaluated the dangerous screw holes in mid-anterior humeral plating using the locking plate and reported that insertion of the fourth to eighth screws carried a risk of iatrogenic radial nerve injury [26]. They found that the incidences of these screws contacting or damaging the radial nerve were 22.2%, 38.9%, 50.0%, 44.4% and 16.7%, respectively. The 4.5 mm narrow DCP is one of the standard plates in anterolateral humeral shaft plating [27] because it provides good stability [28] at a lower price, and is also available in all hospitals, which are notable considerations in developing country. However, to our knowledge, there have been no studies to date evaluating the dangerous screw holes in anterolateral humeral shaft plating using the 4.5 mm narrow DCP.

There were two purposes of this study. The first was to identify the screw holes that are most dangerous in terms of causing an inadvertent radial nerve and/or profunda brachii artery injury while drilling the holes or inserting the screws for the 4.5 mm narrow DCP plate. The second was to measure the distances between the screw tips to the radial nerve and profunda brachii artery in anterolateral humeral plating with this plate. Due to the location of the plate and direction of the screws, we hypothesized that the dangerous screw holes in humeral plating using the 4.5 mm narrow DCP would be the 1st proximal and 2nd proximal screw holes.

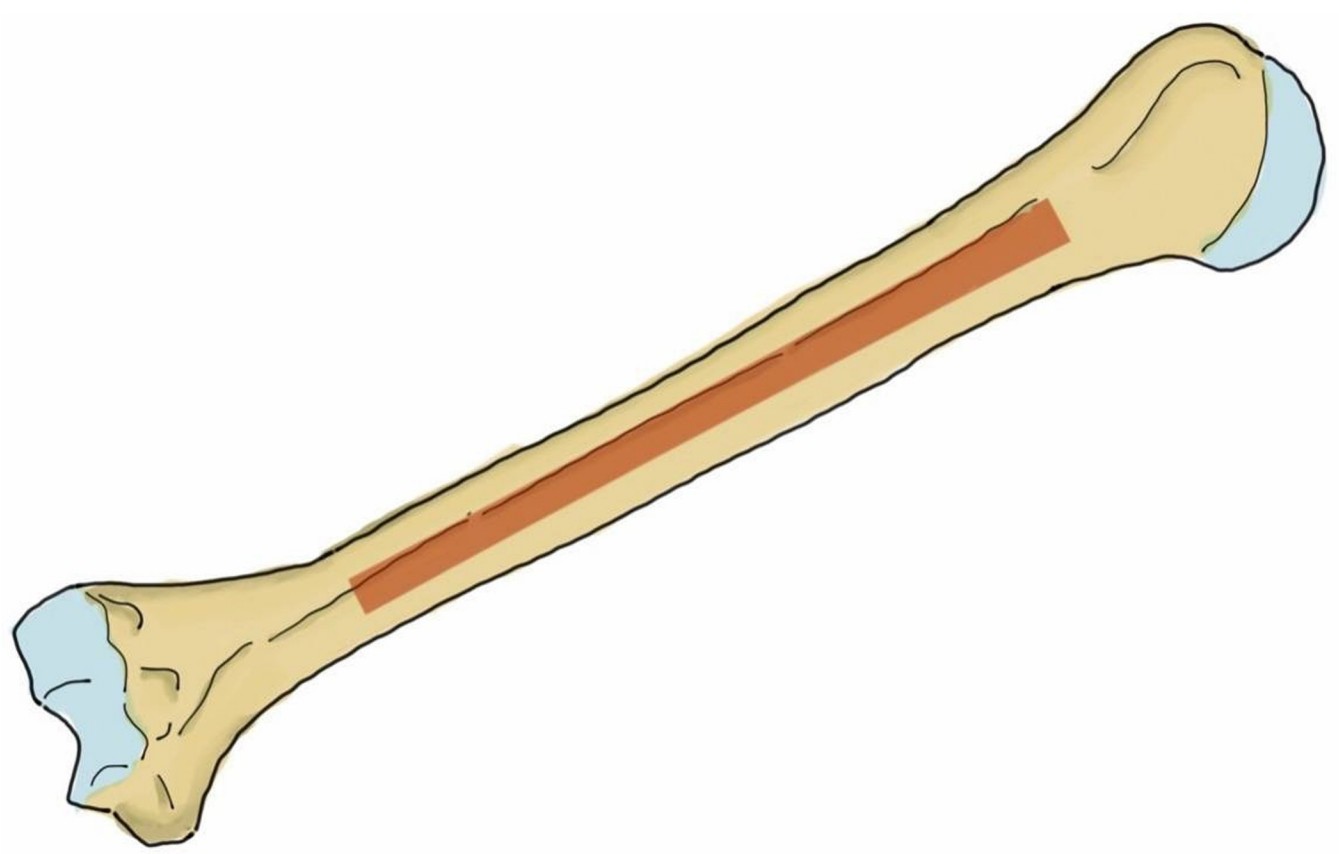

**Fig 2. The mid-anterior aspect of the humerus (brown shaded area) shows the area in which the plate can be applied.**

## Material and methods

18 humeri of 9 full-body fresh-frozen cadavers, 5 males and 4 females, with a mean age at death of 60 ± 10 years, obtained from Prince of Songkla University were used for the study. This study has been approved from the Ethics Committee of Prince of Songkla University (REC 62-434-11-1). The average length of the cadavers was 166 ± 8 cm, and the average humeral lengths in the male and female cadavers were 297.25 ± 2.12 mm and 277.50 ± 1.73 mm, respectively. Each cadaver was thawed at room temperature for 12 hours before the procedures were performed.

For the procedures, each cadaver was first placed in the supine position with 45 degrees of arm abduction and full supination of the forearm. A well- experienced orthopaedic surgeon (15+ years) used the standard anterolateral approach for humeral plating with a ten-hole 4.5 mm narrow DCP. The initial skin incision was made from the tip of the coracoid process to 5 cm proximal to the flexion crease of the elbow. The deltoid muscle was retracted laterally and the pectoralis major retracted medially. The brachialis muscle was split to expose the humeral shaft. A fracture line was set at the midpoint between the tip of the greater tuberosity to the lateral epicondyle. A ten-hole 4.5 mm narrow DCP was contoured to match the anterolateral surface of the humerus, and applied at the anterolateral aspect of the humerus using the fracture line to position the center of the plate. In this step, if the deltoid insertion interfered with the plate application, the deltoid insertion was detached as little as possible to allow the application. When the plate was properly in place, the screw holes were numbered in two directions moving outward from the midpoint of the plate. The hole closest to the midpoint in the

proximal humerus direction was identified as the first proximal screw hole, the next hole in that direction the second proximal screw hole, etc. until the last was identified as the fifth proximal screw hole. For the screw holes on the side toward the distal humerus, the hole closest to the midpoint in the distal direction was identified as the first distal screw hole, the next hole in that direction the second distal screw hole, etc until the last was identified as the fifth distal screw hole. The holes were then drilled and tapped and the cortical screws inserted by hand. To decrease potential bias from the drill and screw directions, a drill sleeve was positioned perpendicular to the plate and centered to the bone by the surgeon and an assistant. After the fixation, the distance from the lateral epicondyle of the humerus to the center of each screw hole was measured along the lateral border of the humeral shaft for calculation of the relative ratios with the entire humeral length (Fig 3). Finally the skin was closed layer by layer.

The cadaver was then turned over to the prone position with 45 degrees of arm abduction and full forearm supination. A posterior incision was made from the posterolateral corner of the acromion process to the tip of the olecranon process by an experienced (10+ years) microneurovascular orthopaedic surgeon. A triceps-splitting approach was done between the long head and the lateral head of the triceps and the radial nerve and profunda brachii artery exposed. In this step, only the medial and lateral borders of the radial nerve and profunda

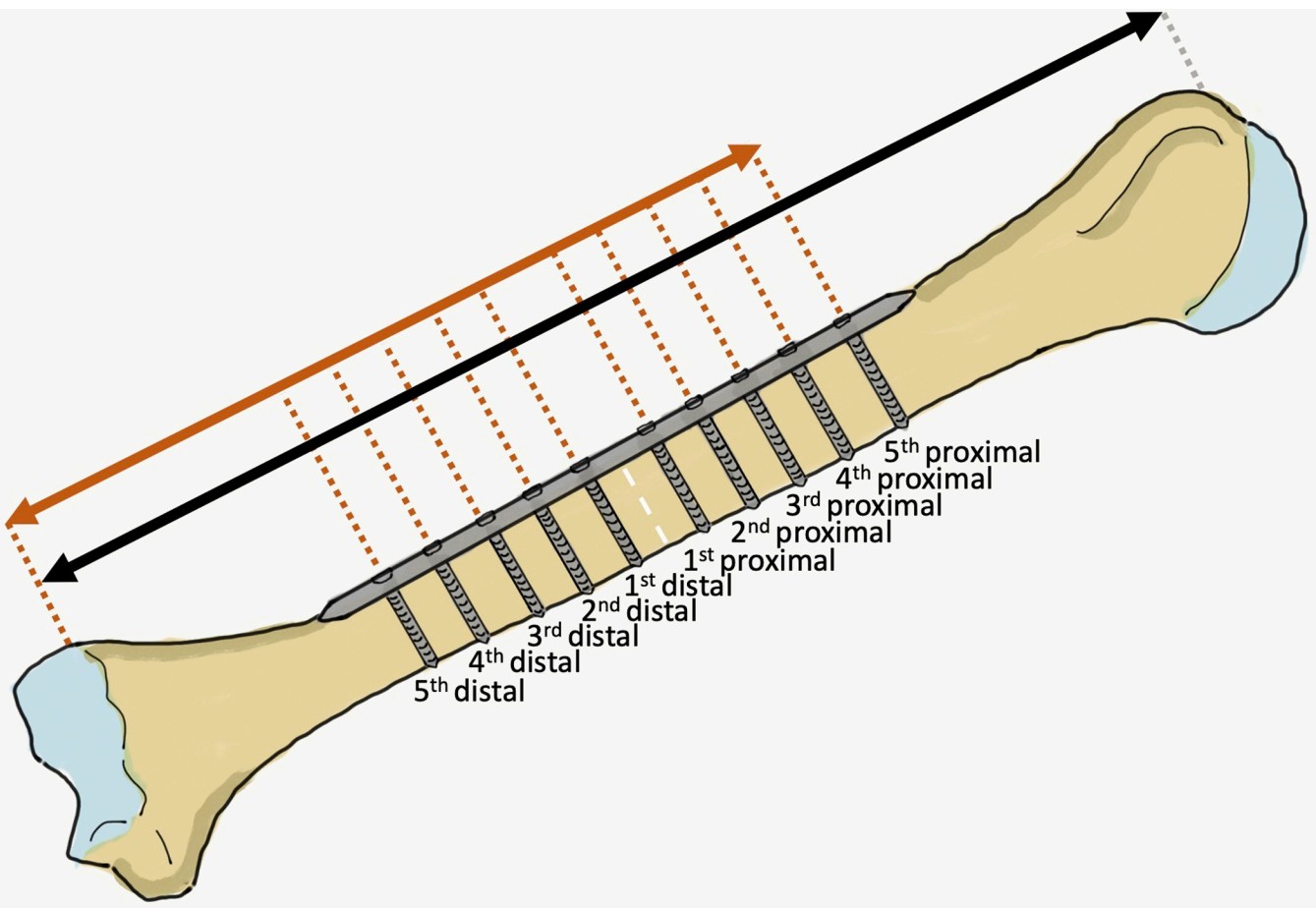

**Fig 3. Anterior view of a right upper arm, with an imaginary fracture line at the midpoint between the tip of the greater tuberosity and the lateral epicondyle (white dotted line).** The distances were measured from the lateral epicondyle to the tip of the greater tuberosity of the humerus (entire humeral length; black arrow) and the center of each screw hole (orange arrow). The relative distance ratio of each screw hole was calculated based on the distance from the lateral epicondyle to each screw hole (orange arrow) and the entire humeral length (black arrow).

brachii artery were exposed to preserve an as near-as-normal relationship of the neurovascular structure to the bone, and the screw holes examined. The distances from the lateral epicondyle of the humerus to the proximal and distal parts of the radial nerve and profunda brachii artery were measured.

In each humerus, the screw holes from which the screw tips were in contact with or had penetrated the radial nerve and/or profunda brachii artery in all cadavers were identified as dangerous screw holes and the screw holes that had penetrated the radial nerve and/or profunda brachii artery in only some cadavers were identified as risky screw holes. The distances from the screw holes in the zone where the radial nerve and profunda brachii artery crossed the humerus to the radial nerve and profunda brachii artery in the longitudinal and closest planes were measured using a Vernier caliper with a precision of 0.001 mm (Insize, Suzhou New District, China) (Fig 4).

To decrease measurement bias, one orthopaedic surgeon measured each distance three times, and the mean ± SD was calculated. The statistical analysis was performed using the R program and "epicalc" package (version 3.4.3; R Foundation for Statistical Computing, Vienna, Austria). Intraobserver reliability was calculated by intraclass correlation coefficients.

## Results

In anterolateral humeral shaft plating with the ten-hole 4.5 mm narrow DCP, the relative distance ratios of the entire humeral length from the lateral epicondyle of the humerus to the 4th, 3rd, 2nd and 1st proximal holes were 0.64, 0.60, 0.56 and 0.52, respectively, while the relative distance ratios of the entire humeral length for the distances from the lateral epicondyle of the humerus to the proximal and distal parts of the radial nerve and profunda brachii artery were

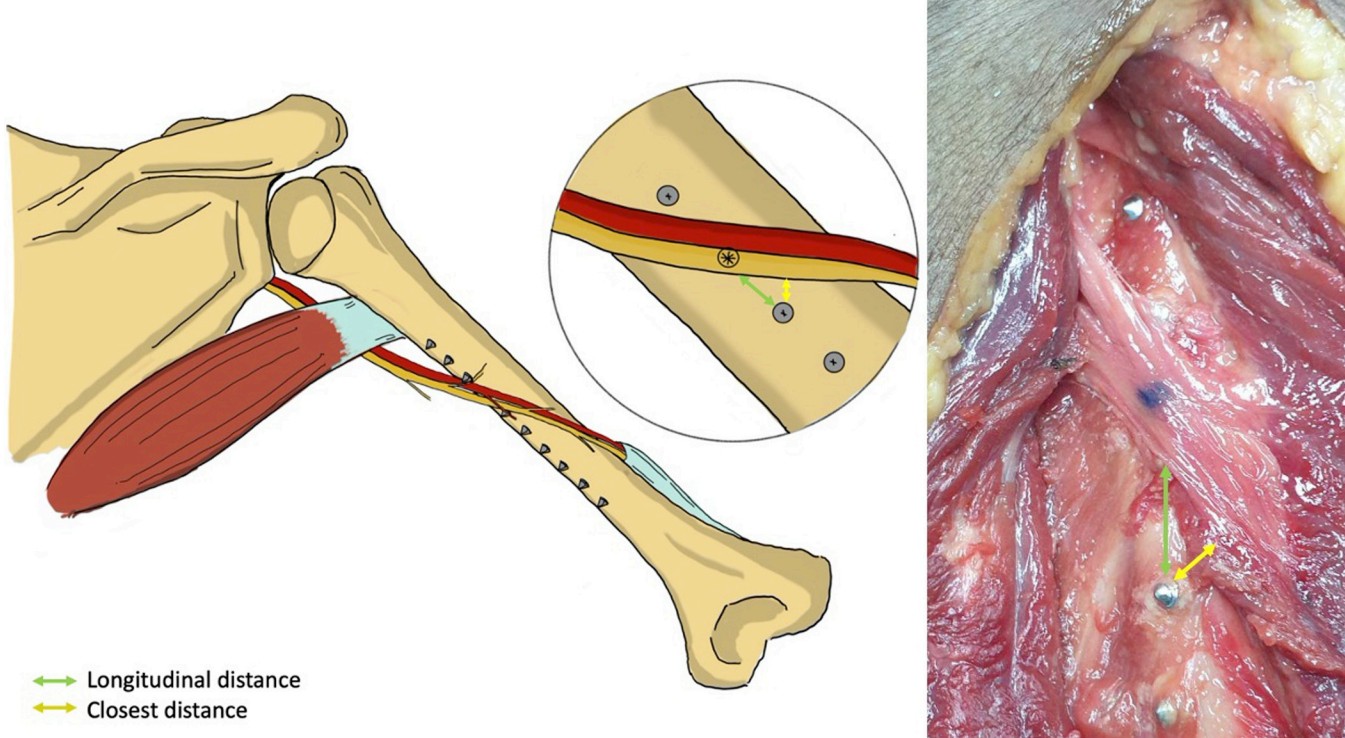

**Fig 4. Posterior view of a right upper arm; the distances from the screw holes to the neurovascular structure were measured in the longitudinal plane (green line) and closest plane (yellow line).**

0.65 and 0.45, respectively. The only dangerous screw hole identified in our study was the 2nd proximal screw hole, while the 4th, 3rd and 1st proximal holes were classified as 'risky', with incidences of iatrogenic radial nerve and/or profunda brachii artery injury of 2/18 (11.11%), 8/18 (44.44%) and 12/18 (66.67%) fixations, respectively (Fig 5).

The average distances from the 4th, 3rd and 1st proximal screw holes to the radial nerve and profunda brachii artery in the longitudinal and closest planes are shown in Table 1. Each distance was measured three times by a single orthopaedic surgeon; the intra-observer reliability ranged from 0.83 to 0.99.

## Discussion

Fixation of a humeral shaft fracture with a 4.5 mm narrow DCP can involve accidental injury to the radial nerve and/or profunda brachii artery because these structures pass very close to the humeral bone in an area where a drill bit or screw can potentially injure them. In our study, we found that the most dangerous screw hole was the 2nd proximal screw hole, with a relative distance ratio of 0.56, for which the surgeon must be very careful not to drill and tap the hole or insert the screw beyond the edge of the far cortex of the bone. The 4th, 3rd and 1st proximal screw holes were identified as 'risky' holes, with varying chances of an iatrogenic neurovascular injury from a bicortical screw.

This study had some limitations. First, the cadavers of different lengths had different humeral lengths, and real patients will of course also have different lengths of humeri. In this

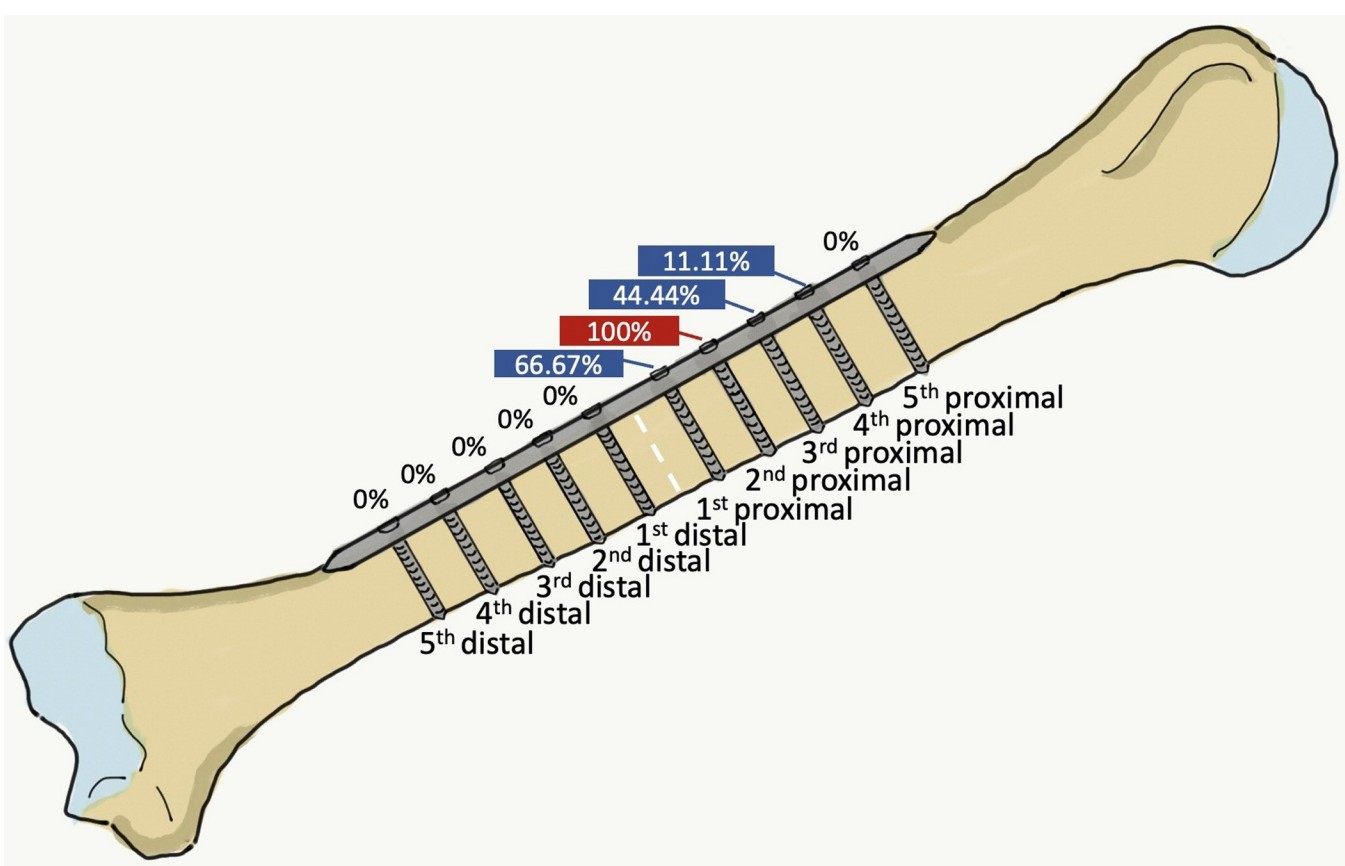

**Fig 5. Anterior view of a right upper arm, showing an imaginary fracture line at the midpoint between the tip of the greater tuberosity and the lateral epicondyle (white dotted line), indicating the risk of iatrogenic radial nerve and/or profunda brachii artery injury of each screw hole.**

**Table 1. The average distances of the 'risky' screw holes to the radial nerve and profunda brachii artery in the longitudinal plane and closest plane (Mean ± SD).**

| Screw hole | Longitudinal plane (mm.) | Closest plane (mm.) |
|---|---|---|
| 4th proximal | 29.19 ± 16.44 | 24.44 ± 14.63 |
| 3rd proximal | 18.69 ± 15.63 | 15.39 ± 13.31 |
| 1st proximal | 12.30 ± 5.13 | 10.75 ± 4.86 |

study, we reported the positions of the "dangerous" and "risky" screw holes that can be applied in the average patient, and we also reported the relative distance ratios of the "dangerous" and "risky" screw holes that can be applied in all sizes of humeri. Second, we tried to keep the normal relationship between the radial nerve/profunda brachii artery and the humerus by exploring only the medial and lateral borders of the radial nerve and profunda brachii artery, but in real patients the anatomical relationship between the radial nerve/profunda brachii artery and the humerus will differ to greater or lesser extents depending on the patient's age, degree of soft tissue injury and the alignment of reduction. Third, we had no way of knowing if there had been any previous injury to the humerus of the cadavers, but cadavers with obvious lesions or deformities of the humerus were excluded. Forth, in actual humeral fractures, the distances from the plate and screws to the radial nerve and profunda brachii artery may be different from the distances in this study, depending on the degree of soft tissue injury and/or reduction alignment.

The radial nerve and profunda brachii artery cross the posterior humerus obliquely from the medial to the lateral sides [6–8]. There is a chance of iatrogenic radial nerve injury in either the anterolateral or the posterior humeral shaft plating, with incidences of between 3.57% and 18.20% [9–11]. Previous studies have reported the incidence of radial nerve palsy in the anterolateral plating was similar to the posterior plating, but significantly higher only in one study [4]. The posterior plating can endanger the radial nerve and/or profunda brachii artery by plate compression or accidental injury during surgical exposure while the anterolateral plating can injure the radial nerve and/or profunda brachii artery while drilling the screw holes and/or inserting the screws [9].

Several studies have evaluated the anatomical location of the radial nerve in relation to different landmarks such as the olecranon process, lateral epicondyle, medial epicondyle, intermuscular septum and/or acromial process [8, 29–31]. The study of Suwannaphisit et al. found that the radial nerve passed the posterior humerus from 130.00 ± 2.07 mm between the upper olecranon and the center of the radial nerve to 122.00 ± 2.33 mm between the lateral epicondyle and the lateral intermuscular septum [30]. In another study, in relation to the epicondyle, the radial nerve passed the posteromedial-to-posterolateral aspect of the humerus between 152 mm and 66 mm from the medial epicondyle [8]. Chou et al. reported the distances from the acromial process to the upper margin and lower margin were 147 ± 21 mm and 195 ± 36 mm, respectively [31]. Additionally, Natsis et al. reported a rare anatomical variation of the radial nerve at the upper humeral level involving an atypical communicating branch between the radial nerve and ulnar nerve, with an incidence was 2.3% [32]. Due to the anatomy and natural variations of the exact location the radial nerve crosses the humerus, there is always a risk of iatrogenic radial nerve injury in anterolateral plating when the screws exit the bone in the area where the radial nerve passes the posterior humerus. Using the results of our study, the surgeon can evaluate the risk of injury for each screw hole separately in their particular situation. In our study, we found that inserting the screw into the 2nd proximal screw hole, which had a relative distance ratio of 0.56, resulted in a 100% chance of iatrogenic radial nerve and/or profunda brachii artery injury. The "risky" screw holes with this plate were the 4th, 3rd and 1st

proximal screw holes, for which the relative distance ratios were 0.64, 0.60 and 0.52, respectively. The results from our study were different from a previous study of Apivatthakakul et al. which reported that the incidence of screws contacting or damaging the radial nerve in the fourth to eighth screw holes of a 4.5 mm narrow LCP were 22.2%, 38.9%, 50.0%, 44.4% and 16.7%, respectively [26]. These differences could be explained by noting the different methods between our study and the Apivatthakakul et al. study, notably the types of plate used, the different directions of the drill bit and screws between a 4.5 mm narrow LCP and a DCP, and the locations and landmarks used when applying the plate.

To avoid the potential of injury with the DCP, we recommend measuring and calculating the relative distance ratios intraoperatively. Inserting a bicortical screw at the $2^{nd}$ proximal screw hole can endanger the radial nerve and/or profunda brachii artery by the tip of the drill bit when drilling the screw hole, the tip of the depth gauge when measuring the length of the screw hole, the tip of the tapping device when tapping the screw hole, or the tip of the cortical screw. To avoid the risk of radial nerve and/or profunda brachii artery injury using the 4.5 mm ten-hole narrow DCP, the surgeon should drill and tap only one side of the cortex and use a unicortical screw. For the other "risky" screw holes, the surgeon should be careful in all steps of drilling the holes and inserting the screws which have a chance of iatrogenic radial nerve and/or profunda brachii artery injury.

## Conclusion

In anterolateral humeral shaft plating with the 4.5 mm narrow DCP using the anterolateral approach, we found that the $2^{nd}$ proximal screw hole with a relative distance ratio of 0.56 carried the highest risk of iatrogenic radial nerve and/or profunda brachii artery injury. We recommend that only a unicortical screw should be used for the $2^{nd}$ proximal screw hole with the 4.5 mm narrow DCP using the anterolateral approach.

## Supporting information

**S1 Data.**
(XLSX)

## Acknowledgments

We thank Weerachai Samai, MD, Head of the Pathology Department of the Faculty of Medicine; Chittipong Tipbunjong, PhD, head of the Anatomy Department of the Faculty of Science, and their assistants for arranging the cadavers; Boonsin Tangtrakulwanich, MD, PhD, of the Department of Orthopedics and Nannapat Pruphetkaew, MSc, of the Epidemiology Unit, Faculty of Medicine, for providing statistical support; and David Patterson of the International Affairs Office of the Faculty of Medicine for his English editing.

## Author Contributions

**Conceptualization:** Chaiwat Chuaychoosakoon, Supatat Chirattikalwong, Tanarat Boonriong.

**Formal analysis:** Chaiwat Chuaychoosakoon, Supatat Chirattikalwong, Wachiraphan Parinyakhup.

**Funding acquisition:** Chaiwat Chuaychoosakoon.

**Investigation:** Sitthiphong Suwannaphisit.

**Methodology:** Supatat Chirattikalwong, Watit Wuttimanop, Wachiraphan Parinyakhup, Sitthiphong Suwannaphisit.

**Project administration:** Chaiwat Chuaychoosakoon.

**Supervision:** Tanarat Boonriong.

**Writing – original draft:** Chaiwat Chuaychoosakoon, Wachiraphan Parinyakhup, Sitthiphong Suwannaphisit.

**Writing – review & editing:** Chaiwat Chuaychoosakoon, Sitthiphong Suwannaphisit.

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
