## [Decision Letter · Decision Letter 0]

9 Oct 2021

PONE-D-21-25577The Risk of Iatrogenic Radial Nerve and/or Profunda Brachii Artery Injury in Anterolateral Humeral Plating using a 4.5 mm Narrow DCP: A Cadaveric StudyPLOS ONE

Dear Dr. SUWANNAPHISIT,

Thank you for submitting your manuscript to PLOS ONE. After careful consideration, we feel that it has merit but does not fully meet PLOS ONE’s publication criteria as it currently stands. Therefore, we invite you to submit a revised version of the manuscript that addresses the points raised during the review process.

Please explain in detail why you prefer an anterolateral approach in fracture types carrying a risk of iatrogenic risk of radial nerve as well as profunda brychii artery insteasd of a dorsal approach, which provides a direct visualisation of nerve and vessels. Refer to any comment of the reviewers.

We look forward to receiving your revised manuscript.

Kind regards,

Hans-Peter Simmen, M.D., Professor of Surgery

Academic Editor

PLOS ONE

Journal Requirements:

2. In the Methods section of the manuscript please ensure that you have specified the full name of the university which provided the Cadavers for the study.

5. Please include a copy of Table 1 which you refer to in your text on page 10.

Reviewers' comments:

Reviewer's Responses to Questions

**Comments to the Author**

1. Is the manuscript technically sound, and do the data support the conclusions?

Reviewer #1: Partly

Reviewer #2: Yes

2. Has the statistical analysis been performed appropriately and rigorously? 

Reviewer #1: Yes

Reviewer #2: I Don't Know

3. Have the authors made all data underlying the findings in their manuscript fully available?

Reviewer #1: No

Reviewer #2: Yes

4. Is the manuscript presented in an intelligible fashion and written in standard English?

Reviewer #1: Yes

Reviewer #2: No

5. Review Comments to the Author

Reviewer #1: Basically the authors have found out, that the "2nd proximal screw" is dangerous (100% hits) with other proximal screws (1/3 and 4) being risky (11-67%). The most proximal and all screws distal to the humerus' middle are not risky. Unfortunately, I was not able to open/view Table 1. The figures are all good, instructive and of good quality.

L82: Dorsal plating with direct visualization of the radial nerve is also a popular technique which needs to be mentioned. I am not so sure that most surgeons prefer anterior plating… (at least not for midshaft fractures and those located more distally)

L88-91: Why is the plate position different for DCP and LCP, respectively? Aren't they placed the same way?

L100: The authors state that the narrow 4.5 DCP is the most commonly used plate for the humerus shaft. However, no references are provided.... I strongly disagree with the authors. I do not think that is true for Europe and North America and other developed countries.

L 232 Different directions of the drill bit? In LCP the locking screws are inserted in a perpendicular manner as well. I don't see a difference here? Please explain.

I would like to have studies discussed which have investigated the anatomical variations of the radial nerve in relation to the humeral bone. This aspect is totally neglected in the discussion. Furthermore, results and complications (regarding the radial nerve and accompanying artery) of anterior humeral plating should be reviewed and discussed with the findings of your study. Only one similar study is (Apivatthakakul et al) is mentioned.

I would suggest a more thorough and critical review of the literature regarding the variation of the radial nerve and the accompanying vessels, other experimental studies and results and complications of clinical studies with anterior plating.

Reviewer #2: Line 50: "A hypothetical fracture line was marked at the midpoint of each humerus." Can you provider some data why you choose the fracture line there?

Line 79 Introduction: This part is to short. You should give some feedback information about the percentage of humerus fractures and numbers about prim. and sec. nerve damage.

Line 80: Can you make some comments about the posterior approach for plating and the percentage of nerve damage from each approach?

Line 92: re-phrase please

105-111: please rephrase this very long sentence.

Line 124: can you be more specific what "experienced means with a number of years in experience

Line 130: see comment 50

Line 154: see comment number 124:

Line 215: rephrase in a more scientific way.

Line 223: Important discussion/limitations as well is the fact, that the holes this close to the fracture, according the AO-Guidelines, are normally left empty. Please add this as a discussion point.

Line 245: I don’t fully see the connection between this two studies. And according to the AO cerclage wiring are not recommended at the humerus shaft.

Line 256: What about using a different approach like the posterior one? There are studies out that this approach when used primarily, is also a safe and sound way for plaiting-.

General information:

It would be interesting to know, how the screws where inserted (by hand or with the drill) and if there would be a lesser chance with nerve injurie by hand use.

I don’t have access to the statistical data.

6. PLOS authors have the option to publish the peer review history of their article (what does this mean?). If published, this will include your full peer review and any attached files.

Reviewer #1: No

Reviewer #2: No

---

## [Author Response · Author response to Decision Letter 0]

2 Nov 2021

Journal Requirements:

ANS This manuscript has been revised following the PLOS ONE template.

2. In the Methods section of the manuscript please ensure that you have specified the full name of the university which provided the Cadavers for the study.

ANS The full name of the university has been added in the manuscript on lines 126-127.

ANS The data has been uploaded in the supporting information files.

ANS This ethic of this study has been approved from the Faculty of Medicine of Prince of Songkla University.

5. Please include a copy of Table 1 which you refer to in your text on page 10.

ANS Table 1 has been added in the manuscript.

Editorial comment:

Please explain in detail why you prefer an anterolateral approach in fracture types carrying a risk of iatrogenic risk of radial nerve as well as profunda brachii artery insteasd of a dorsal approach, which provides a direct visualisation of nerve and vessels.

ANS The anterolateral plating is suitable for a fracture in the area from the proximal to the mid-shaft of the humerus, while the posterior plating is more appropriate for fixations between the mid-shaft and distal humerus. In mid-shaft humeral fixation, the surgeon can use either the anterolateral or posterior plating. In a national survey, both anterolateral and posterior approaches were popular in treating a humeral shaft fracture. There were good to excellent outcomes in both approaches. The complication rates of iatrogenic radial nerve palsy in most studies were similar between the two locations of the plate with incidences of 3.57 – 14.2% in the anterolateral plating and 16.67 – 18.18% using the posterior plating. In contrast, there was one study which reported that the incidence of iatrogenic radial nerve palsy in the anterolateral approach was higher than in the posterior approach. In that study, 46 of 707 humeral shaft fixations had iatrogenic radial nerve injury, compared to 37 cases from the anterolateral approach and 2 cases from the posterior approach. When applying the plate at the anterolateral surface of the bone, there was a risk of iatrogenic radial nerve injury in the steps of drilling a screw hole and/or inserting a screw. In the posterior plating, there was a risk in the step of identifying the radial nerve and profunda brachii artery as they pass across the mid-posterior aspect of the humerus. 

A humerus fracture can be caused by a high-energy mechanism and in such cases, the patient will often have associated injuries precluding surgery in the prone or lateral decubitus position. In this study, we used the anterolateral approach because this approach can be done in the supine position, which is suitable for most patients. The posterior approach can be used in either the prone or lateral decubitus positions, although there are contraindications in cases of spinal instability or patients at risk of spinal instability (eg, rheumatoid arthritis), unstable fractures (especially facial and pelvic), anterior burns, chest tubes, and open wounds, shock, pregnancy, recent tracheal surgery, and high intracranial pressure. 

Comments to the Author

Review Comments to the Author

Reviewer #1: Basically the authors have found out, that the "2nd proximal screw" is dangerous (100% hits) with other proximal screws (1/3 and 4) being risky (11-67%). The most proximal and all screws distal to the humerus' middle are not risky. Unfortunately, I was not able to open/view Table 1. The figures are all good, instructive and of good quality.

ANS I am sorry you had trouble with this Table. I have carefully checked and re-attached Table 1 in the re-submission. 

L82: Dorsal plating with direct visualization of the radial nerve is also a popular technique which needs to be mentioned. I am not so sure that most surgeons prefer anterior plating… (at least not for midshaft fractures and those located more distally)

ANS The posterior approach has been mentioned following your suggestion and the sentence of surgeon preferences has been revised on lines 66-69 and 96-100. The surgeon can use either the anterolateral or posterior plating in a mid-shaft humeral fracture. In a national survey of Ahad et al, both anterolateral and posterior plating were popular in treating a humeral shaft fracture. There were good to excellent outcomes in both approaches. To find the radial nerve in the posterior approach, there were many studies reported the exact soft tissue and bony landmarks to identify the nerve. In contrast, to our knowledge only one study has evaluated the risk of injury in mid-anterior humeral plating, but the results of this study could not be used in anterolateral humeral plating because the direction of the screws is different. It would be beneficial for the surgeon to be able to avoid potential injury to the nerve by knowing the “dangerous” screw holes.

• Ahad A, Haque A, Armstrong A, Modi A, Pandey R, Singh HP. The management of displaced humeral shaft fractures – A survey of UK shoulder and elbow surgeons. Shoulder Elb. 2021;0(0):1-6. doi:10.1177/1758573220986940

L88-91: Why is the plate position different for DCP and LCP, respectively? Aren't they placed the same way?

ANS The plate position is different between the 4.5 mm narrow DCP and the LCP. To provide stability for the DCP, the surgeon should apply the plate at a flat surface which is the anterolateral aspect of the humerus, while it is not necessary to apply the LCP at a flat surface. The LCP can be applied at either the mid-anterior surface or the anterolateral surface of the humerus, as described on lines 84-87. To date, there have been no studies evaluating the risk of radial nerve injury when applying the plate at the anterolateral aspect of the humerus, which is the most common plate location in open humeral shaft fixation.

L100: The authors state that the narrow 4.5 DCP is the most commonly used plate for the humerus shaft. However, no references are provided.... I strongly disagree with the authors. I do not think that is true for Europe and North America and other developed countries.

ANS Thank you for your comment. The phrase of “most commonly used” has been changed to “one of the standard plates” on line 105. As below references, the 4.5 mm narrow DCP is one of the standard plates used in humeral shaft fixation and biomechanical testing between the DCP and LCP has found no statistically significant differences in outcomes in comminuted fractures with good bone quality. 

• Gallusser N, Barimani B, Vauclair F. Humeral shaft fractures. EFORT Open Rev. 2021;6(1):24–34. 

• O’Toole R V., Andersen RC, Vesnovsky O, Alexander M, Topoleski LDT, Nascone JW, et al. Are locking screws advantageous with plate fixation of humeral shaft fractures? a biomechanical analysis of synthetic and cadaveric bone. J Orthop Trauma. 2008;22(10):709–15. 

L 232 Different directions of the drill bit? In LCP the locking screws are inserted in a perpendicular manner as well. I don't see a difference here? Please explain.

ANS The location where the plate is applied is the difference. The 4.5 mm narrow DCP is applied at the anterolateral surface of the humerus while the 4.5 mm narrow LCP applied at either the mid-anterior or anterolateral aspect of the humerus on lines 84-87. There has been, to our knowledge, only one study which evaluated plates applied at the mid-anterior aspect of the humerus, and no studies which have evaluated plates applied at the most common location (anterolateral aspect of the humerus). The screw direction of the 4.5 mm narrow DCP applied at the anterolateral aspect of the humerus projects to the posterolateral area while the screw direction of the 4.5 mm narrow LCP applied at the mid-anterior aspect of the humerus projects to the mid-posterior area.

I would like to have studies discussed which have investigated the anatomical variations of the radial nerve in relation to the humeral bone. This aspect is totally neglected in the discussion. Furthermore, results and complications (regarding the radial nerve and accompanying artery) of anterior humeral plating should be reviewed and discussed with the findings of your study. Only one similar study is (Apivatthakakul et al) is mentioned.

ANS Some discussion of anatomical variations of the radial nerve has been added in the Discussion on lines 260-274. We discussed the results of this study by comparing it with the study of Apivatthakakul et al. because this study seems to be the only study which evaluated the risk of injury based on the hole positions in mid-anterior humeral plating.

I would suggest a more thorough and critical review of the literature regarding the variation of the radial nerve and the accompanying vessels, other experimental studies and results and complications of clinical studies with anterior plating.

ANS Thank you for your suggestion. As noted above, we have added some discussion on this point on lines 271-274.

Reviewer #2: 

Line 50: "A hypothetical fracture line was marked at the midpoint of each humerus." Can you provider some data why you choose the fracture line there?

ANS In humeral shaft fractures, the most common fracture location is the middle third of the shaft of the humerus (60%) (as mentioned in the paper referenced below) on lines 49-50, so we located our hypothetical fracture here. For fractures in different areas, the surgeon can use the relative distance ratios we calculated to determine the risk of iatrogenic radial nerve injury. 

• Ahad A, Haque A, Armstrong A, Modi A, Pandey R, Singh HP. The management of displaced humeral shaft fractures – A survey of UK shoulder and elbow surgeons. Shoulder Elb. 2021;0(0):1-6. doi:10.1177/1758573220986940

Line 79 Introduction: This part is too short. You should give some feedback information about the percentage of humerus fractures and numbers about prim. and sec. nerve damage.

ANS We have added the requested information on lines 49-58 and 72-79.

Line 80: Can you make some comments about the posterior approach for plating and the percentage of nerve damage from each approach?

ANS Most previous studies reported that the percentage of iatrogenic radial nerve palsy was not statistically different between anterolateral and posterior humeral shaft plating. There has been only one study which found iatrogenic radial nerve injury using the anterolateral plating was higher than in the posterior approach, as mentioned earlier, which reported injuries in 37 of 39 cases from anterolateral plating and 2 of 39 cases from posterior plating on lines 77-79.

Line 92: re-phrase please

ANS This sentence has been revised following your suggestion on lines 50-57.

105-111: please rephrase this very long sentence.

ANS This sentence has been revised following your comment on lines 115-123.

Line 124: can you be more specific what "experienced means with a number of years in experience

ANS The orthopaedist had more than 10 years experience in micro-neurovascular surgery, which has been added in the manuscript on line 173.

Line 215: rephrase in a more scientific way.

ANS This sentence has been revised following the suggestion on lines 243-245.

Line 223: Important discussion/limitations as well is the fact, that the holes this close to the fracture, according the AO-Guidelines, are normally left empty. Please add this as a discussion point.

ANS Thank you for your suggestion. However, we have checked the most recent AO guideline ( https://surgeryreference.aofoundation.org/orthopedic-trauma/adult-trauma/humeral-shaft/simple-fracture-transverse-less-than30/orif-compression-plating#plate-fixation ), which suggests inserting a screw in a pre-drilled hole in fragment close to the fracture and inserting an eccentrical screw into the other fragment near the fracture.

Line 245: I don’t fully see the connection between this two studies. And according to the AO cerclage wiring are not recommended at the humerus shaft.

ANS This paragraph has been removed from the Discussion.

Line 256: What about using a different approach like the posterior one? There are studies out that this approach when used primarily, is also a safe and sound way for plating

ANS There have been several studies which have evaluated the risk of iatrogenic radial nerve injury between the anterolateral and posterior approaches. The incidences were not statistically significantly different between these approaches.

The radial nerve must be identified when using the posterior approach. In a national survey, some surgeons preferred to use the anterolateral approach in humeral shaft plating which does not require pre-identification of the radial nerve and this approach can be done in all patient situations, while the posterior approach can be done in patients in the prone or lateral decubitus positions.

General information:

It would be interesting to know, how the screws where inserted (by hand or with the drill) and if there would be a lesser chance with nerve injurie by hand use.

ANS In this study, we inserted all screws by hand, as described on line 155. We were unable to find any studies comparing safety between hand-inserted and drill-inserted screws. However, if the surgeon is inserting a screw in any danger area, inserting by hand is generally accepted as safer.

---

## [Editor Report · Decision Letter 1]

10 Nov 2021

The Risk of Iatrogenic Radial Nerve and/or Profunda Brachii Artery Injury in Anterolateral Humeral Plating using a 4.5 mm Narrow DCP: A Cadaveric Study

PONE-D-21-25577R1

Dear Dr. SUWANNAPHISIT,

We’re pleased to inform you that your manuscript has been judged scientifically suitable for publication and will be formally accepted for publication once it meets all outstanding technical requirements.

Kind regards,

Hans-Peter Simmen, M.D., Professor of Surgery

Academic Editor

PLOS ONE
---

## [Editor Report · Acceptance letter]

18 Nov 2021

PONE-D-21-25577R1 

The Risk of Iatrogenic Radial Nerve and/or Profunda Brachii Artery Injury in Anterolateral Humeral Plating Using a 4.5 mm Narrow DCP: A Cadaveric Study 

Dear Dr. Suwannaphisit:

I'm pleased to inform you that your manuscript has been deemed suitable for publication in PLOS ONE. Congratulations! Your manuscript is now with our production department. 

Kind regards, 

on behalf of

Dr. Hans-Peter Simmen 

Academic Editor

PLOS ONE